# Dental Metal Matrix Composites: The Effects of the Addition of Titanium Nanoparticle Particles on Dental Amalgam

**DOI:** 10.3390/ma17071662

**Published:** 2024-04-04

**Authors:** Ryan Moxon, Zhigang Xu, Felix Tettey, Ikenna Chris-Okoro, Dhananjay Kumar

**Affiliations:** 1Department of Mechanical Engineering, North Carolina A & T State University, Greensboro, NC 27411, USA; 2Department of Chemical, Biological and Bioengineering, North Carolina A & T State University, Greensboro, NC 27411, USA; 3Department of Industrial and Systems Engineering, North Carolina A & T State University, Greensboro, NC 27411, USA

**Keywords:** mercury vapor concentration, γ-1 phase, γ-2 phase, XRD, SEM/EDS, microstructural mechanism, Vickers hardness, dental amalgams, nanoparticles

## Abstract

Dental amalgams have been used by dentists for the restoration of posterior human teeth. However, there have been concerns about the release of mercury from amalgams into the oral cavity. The objective of the present research is to study the effect of titanium (Ti) nanoparticles on the microstructural mechanism of the release of mercury vapor in two commonly used brands of dental amalgam (the Dispersalloy: 11.8% Cu; the Sybralloy: 33% Cu). Ti powder was added to both the Dispersalloy and the Sybralloy in increments of 10 mg up to 80 mg. The addition of Ti powder to both brands of dental amalgam has been found to result in a considerable decrease in Hg vapor release. The decrease in the Hg vapor release due to Ti addition has been explained by the formation of strong Hg–Ti covalent bonds, which reduce the availability of Hg atoms for evaporation. The Ti atoms in excess of the solubility limit of Ti in Hg reside in the grain boundaries, which also reduces the evaporation of Hg from the amalgam. The binding of Hg with Ti via a strong covalent bond also results in a significant improvement in mechanical properties such as Vickers hardness.

## 1. Introduction

Dental amalgam material has been a subject of controversy among the dental community worldwide due to the release of mercury (Hg) vapor from the material when placed in human teeth during and subsequent to the placement of the material for clinical restorations of posterior human teeth [1,2,3]. Such material has been utilized by dentists in the United States for the restoration of posterior human teeth since 1833, with additional altercations in material compositions [4,5,6,7]. The purpose of the research was to determine and investigate the effects of the addition of titanium powder to dental amalgam constituents and determine how, if any, the mercury vapor released from the dental amalgam is affected by the incremental addition of titanium powder to the alloy. Attempts have been made to reduce the amount of mercury vapor released from dental amalgam, and this research is directed at providing some insights in that direction [8,9,10]. Titanium is a biocompatible material used in the manufacturing of both dental and medical implants. The material is classified as anti-corrosive and is adaptable to human tissue, thus rendering a good immune response when placed as implants in humans [11,12,13].

The Dispersalloy (11.8% Cu) is considered a moderately high copper dental amalgam that releases high levels of mercury vapor initially when condensed in the tooth cavity [14,15]. The alloy composition of the material is approximately 50 wt.% to 57 wt.% silver, 24 wt.% to 26 wt.% tin, 6 wt.% to 10 wt.% copper, and 1.5 wt.% zinc. However, the percentages of metal constituents in the Dispersalloy (11.8% Cu) are not conclusive to the material in question [16]. The Dispersalloy varies in composition depending on the manufacturer of the amalgam. 

The Sybralloy (33 % Cu) is classified as a high copper amalgam alloy due to its copper percent in the alloy composition. The high-copper alloy consists of the following elemental composition: silver (40 wt.% to 60 wt.%), tin (12 wt.% to 30 wt.%), copper (10 wt.% to 33 wt.%), zinc (1 wt.% to 2 wt.%), indium (1 wt.% to 4.4 wt.%) [17,18]. The silver in the alloy renders a very high external surface shine during the polishing of the filling material by the dentist subsequent to its placement in the tooth-prepared cavity [18,19]. The tin creates the setting contraction and gives the alloy its malleability properties, thus increasing the setting time of the material [20,21,22]. Copper minimizes corrosion, tarnishing, creep, and the formation of γ-2 phases, as well as improving strength and marginal leakages [22,23,24]. Zinc is the scavenger element. It reduces the oxidation of other elements present within the alloy [25,26]. The Sybralloy (33 wt.%) amalgam brand is a category of dental amalgam belonging to the Admixed High Copper Alloy. During trituration, tin diffuses to the surface of Ag-Cu particles and reacts with copper to form Cu_6_Sn_5_ (eta-phase) around the unconsumed Ag-Cu particles [27]. 

The reaction phases formed by the trituration of dental amalgam are as follows:
The γ-phase (Ag_3_Sn), γ (the strongest phase). It is a closed-pack hexagonal structure, as determined by Nial, Almin, and Westgren (1931). Moreover, Murphy (1926) studied the equilibrium conditions of the silver–tin binary system and confirmed that the γ-phase is of the hexagonal close-pack structure [28,29,30].
The γ-1 phase (Ag_2_Hg_3_), γ_1_-phase (dominant phase in the set amalgam). The unit cell of this γ-1 phase is of the cubic crystal structure, as determined by G. V. Black in 1895, and appears to exist in the (112) plane [31].The γ-2 phase (Sn_8_Hg), γ_2_ the weakest and most corrosive phase. It has a hexagonal crystal structure [31,32,33].The beta phase (Ag_5_Sn), β-phase, is formed by the transformation of the mercury-rich γ-1 phase (Ag_2_Hg_3_). The first phase is often formed in the form of dendrites, then branching and tree-like crystals embedded within the γ-phase matrix [34,35].The eta-phase (Cu_6_Sn) is formed by the unreacted eta-phase (Cu_3_Sn). This is a hexagonal unit crystal structure [30]. In the eta-phase (Cu_6_Sn), copper is added to the alloy by means of the silver–copper eutectic or as the eta-phase (Cu_6_Sn) [36].

The corrosive or unwanted phase is the γ_2_-phase, which contributes to the major release of mercury vapor, marginal leakage, and corrosion of the material. Another stable phase formed is the eta-phase (Cu_6_Sn_5_), which is noted as one of the strongest phases formed in the mixture, with the γ-1 phase being the strongest phase [37,38,39].

As the amalgam ages, the material undergoes high creep, increased micro-expansion, and slight oxidation, therefore enabling some minute amount of corrosion to occur. When teeth are restored with an amalgam over a long period of time, about 10% to 15% of tooth fillings will crack/fracture due to the micro-expansion of the material within the tooth cavity [39]. The delayed expansion of the amalgam is caused by the zinc properties, the development of hydrogen gas from the oral environment, and moisture within the material, further enabling the material to exist in a plastic form in the oral environment [40]. Both the silver and copper contents in the alloy increase the strength and expansion equally, with the copper functioning also to decrease creep. The weight percent tin in the eta-phase (Cu_3_Sn) of the amalgam has been computed as 37.4, and that of the γ-phase (Ag_3_Sn) is 25.6 [41]. The amalgam must not be over-triturated since brittleness will be induced within the silver-rich beta phase resulting from the dominant γ-1 phase (Ag_2_Hg_3_) [42]. The excess vibrations will tend to separate the soft γ-1 matrix phase from the unreacted γ-phase (Ag_3_Sn). After amalgamation (the mixing of Ag-Sn-Cu with elemental mercury), several unreacted γ-phases remain, and over-trituration leads to the further decomposition of the γ-1 phase, resulting in the fracture of the amalgam alloy [43].

When the elemental mercury is triturated with the other metal elements, the mercury is absorbed within the metal particles and dissolves first on the periphery of the metal conglomeration. Due to aging, the reaction products developed a protective layer of oxide on the external circumferential surface of the metal constituents. Silver and tin become saturated and continue to dissolve in the elemental mercury in slow progression during the setting [44]. During the trituration process, the mercury diffuses into the silver–tin (Ag_3_Sn) phase, the γ particles, resulting in the dissolution of both silver and tin into the mercury to a limited extent. The particle sizes become smaller, and the silver, being less soluble in mercury than tin, silver precipitates out first as the γ_1_-phase (Ag_2_Hg_3_), followed by tin–mercury, known as the γ_2_-phase (Sn_8_Hg) [45]. Both the γ_1_-(Ag_2_Hg_3_), having a body-cubic center unit cell structure, and the γ_2_-phase (Sn_8_Hg), with a hexagonal unit structure, continue to precipitate during the first 24 h subsequent to trituration, when the material reaches its maximum strength [28,31,34,46,47].

## 2. Materials and Methods

Samples of dental amalgam utilized during this research were the Dispersalloy (11% copper) and the Sybralloy (33% copper), as shown in Table 1, with each capsule having spills of 600 mg. The Dispersalloy (11 % copper) alloy samples were obtained from Darby Dental Supply (Jericho, NY, USA). Equipment used during this research was the Jerome J505 Arizona Mercury Vapor Analyzer (Arizona Instrument LLC, Chandler, AZ, USA) [48] and the Aphrodite High Speed Digital Amalgamator HL-AH G8 (Aphrodite Pharmaceuticals LLC, Santa Barbara, CA, USA). The Darby D-phase 11 amalgam alloy is a self-activating condensable alloy with a negligible γ-2 phase. The material is known for its marginal integrity, corrosive resistance, and low creep of about 0.2%. After about six hours of condensation, it is known to exhibit a compressive strength of about 23,000 psi. The average regular working time is about three minutes, with a set time of about five minutes.

The Sybralloy (33% copper) amalgam capsules were obtained from Kerr Dental Supply (Kerr Dental, Brea, CA, USA). The alloy is a regular set of 600 mg spills, excellent clinical properties, high silver content, and spherical particles in its alloy composition. The amalgam alloys were mixed with elemental mercury in the proportional ratio of 1 to 1 weight percent ratio. This brand of amalgam was selected for the research due to its high copper content and remarkable strength under compression.

The measuring equipment used for the research was the Jerome J505 Mercury Vapor Analyzer (Arizona Instrument, Chandler, AZ, USA). The Jerome J505 Mercury Vapor Analyzer works on the principle of drawing a measured volume of saturated air with mercury atoms through a built-in internal pump, which creates a vacuum suction within the instruments. The Jerome instrument has a gold metallic strip of metal located inside the air storage chamber of the Jerome instrument. Since gold has a very strong affinity for the attraction of mercury atoms, the mercury atoms accumulate into the golden metallic strip as the saturated air with mercury vapor flows over the golden metal strip. The Jerome instrument measures the density of mercury atoms accumulated on the golden metal strip and displays the quantity of mercury atoms on the screen of the instruments in terms of mass per unit volume of mercury atoms (concentration in kg/m^3^, ng/cm^3^, and µg/cm^3^). Both brands of dental amalgam, the Dispersalloy and Sybralloy capsules amalgam, were utilized for this research.

Titanium powder (99.99% purity) was obtained from Thermo Fisher Scientific (Alfa Aesar), Waltham, MA, USA. The Ti powder (44 µm particle size), consisting of fine particles, was weighed in 10 mg, 20 mg, and 30 mg, was inserted in the powdered metallic alloy of the amalgam capsule and was tightly enclosed. The capsules, each containing incremental measurement of titanium, were placed between the prongs of the Zenith Mercury Amalgamator and triturated at a maximum speed of 48,000 rpm for 15 s.

The titanium powdered particles were weighed in increments of 10 mg, each using the Ohaus PX323/E Pioneer Precision Balance Scale. Subsequent to the measurements of titanium, the titanium powdered particles were mixed with the metallic constituents of the amalgam prior to trituration. Amalgam samples were then prepared, polished, and examined using the methods depicted in this research. All samples were prepared under standard conditions, including amalgamation time and trituration time, and all data were obtained under the same conditions.

The capsules contained the amalgam alloy powder and elemental mercury, with the liquid mercury in separate compartments and enclosed within the capsule. The alloy was doped with an incremental measured quantity of titanium by inserting the titanium into the capsule together with the metallic powder. The capsule was then closed tightly and placed between the prongs of the Aphrodite High Speed Digital Amalgamator HL-AH G8 to be triturated at a fifteen-second duration per cycle of mixing. During the mixing process, it is significant that the alloy must not be over-triturated due to excessive vibrational stresses that may be established at the grain boundaries, thus contributing to areas of weakness and fracture between the grain boundaries. Materials that are over-stressed will possibly result in excessive fracture during and subsequent to condensation. The Ag_2_–Hg_3_ combination is the γ_1_-phase, which is the dominant phase formed subsequent to trituration and from which the γ-2 phase is formed [13].

Prior to initiating the measurement, the Mercury Vapor Analyzer was warmed up for about 10 min, and a test sample reading was obtained to verify the accuracy of the measurements. Amalgam capsules were then condensed and prepared in uniform sizes of 10 mm diameter by 4 mm thickness. The samples were inserted into a Stony Lab 250 mL borosilicate glass (Nesconset, NY, USA). A 12-inch length of plastic tubing with a diameter of 4 mm was attached to the Stony Lab glass, while the other end was attached to the J505 instrument. At the onset of operations, saturated ambient air containing mercury atoms was drawn through a 12-inch plastic tube with a diameter of 4 mm attached to the 250 mL flask. Saturated air samples containing mercury atoms were drawn into the instrument by means of a built-in electrical pump located inside the instrument. The normal flow rate of saturated air is 1 Liter/min. The sample air then flowed through a scrubber filter and directly into the sample cell located inside the instrument; the entrance of the sample cell was controlled by a one-way valve to prevent back-flow.

To measure the evenness of the distribution of Ti and other various constituents and to estimate their relative abundance in the samples, a Hitachi SU8000 (Hitachi USA, Pittsburgh, PA, USA) scanning electron microscopy (SEM) and energy-dispersive X-ray spectroscopy (EDS) were used on samples containing 10 mg, 20 mg, and 30 mg titanium-doped particles.

To understand the phase purity and orientation of the Dispersalloy, X-ray diffraction analysis was performed using the Bruker D8 Machine (Billerica, MA, USA).

Six weeks subsequent to the sample preparation, the hardness of each sample was measured using the MicroVickers Hardness Tester, Model M-400-H1 (Mitutoyo, Kawasaki, Japan). Vickers hardness performance tests were performed on all samples. The hardness number was determined from the formula, as shown in Equation (1). All the measurements were obtained at room temperature during the research experiment (temperature between 20 and 30 °C at 1 atm).

To ensure the validity and reproducibility of the data presented herein, samples were measured under identical conditions at least four times. This enabled the establishment of the error bar for a particular data point.

The Vickers hardness equation is the following:(1)$$VH=\frac{2\times F\sin\left(\frac{\beta }{2}\right)}{g\times D^{2}}=\frac{0.0018549\times P}{D^{2}}$$
where the following is true:

*P* = Applied load used for the indentation of each sample (200 g).

*D* = Length of the diagonal distance of the diamond shape indentation.

*G* = Acceleration due to gravity.

*Β* = Angle of the indenter.

## 3. Results and Discussion

Table 2 and Table 3 indicate that when the titanium nanoparticles were added to both the Dispersalloy (11.8% Cu) amalgam and the Sybralloy (33% Cu) amalgam, the mercury vapor released from each sample decreased at each successive addition of titanium with the Table 4 illustrating a cumulative of the various dental amalgams (with their various Ti content) and the quantity of mercury vapor release in 400secs. Such phenomena are a direct result of the replacement or substitution of the titanium for the mercury in the γ-2 phase and, to a lesser extent, created a substitution between the mercury and titanium atoms in the γ-1 (Ag_2_Hg_3_) phase. The increased content of titanium in the alloy created a change in the material properties, such as an increase in the hardness of both materials. The stoichiometric ratio and form of arrangements of titanium atoms in the phases are not fully understood [43], which necessitated the SEM/EDS analysis conducted much later.

The plot indicated in Figure 1a,b shows the mercury vapor released from eight of the Dispersalloy (11.8% Cu) amalgam samples, beginning with the sample having the lowest amount of titanium (10 mg titanium), with increments of the addition of 20 mg, 30 mg, and 40 mg titanium to each of the other samples. As can be observed from Figure 1, the pattern of Hg vapor released shows a relatively linear drop-in Hg vapor for each of the Dispersalloy amalgam samples. The 11.8% copper within the amalgam matrix of the Dispersalloy was inserted as a constituent of the amalgam by the manufacturer. The measurement of Hg vapor released from each sample was obtained over a time range of 400 s. Such a chosen time range is of no particular significance since any time range could be equally appropriate for the Hg measurement. Based on prior measurements of Hg vapor released from dental amalgam, the pattern of released Hg vapor beyond 400 s will continue in a similar formation to what is observed in the plot of Figure 1.

Figure 1 shows that without the addition of titanium nanoparticles to the amalgam alloy, the material releases a significantly high amount of Hg vapor, as determined from the above plot labeled in black. With the addition of 10 mg titanium (red line), 20 mg titanium (blue line), 30 mg titanium (green line) & without any Titanium content (black line), the mercury vapor decreased due to the displacement of the tin in the γ-2 phase by the titanium particles, which decreased the Hg vapor released from the γ-2 phase. The Dispersalloy (11.8% Cu) is classified as a moderately high-copper amalgam, ≥6% copper, and forms a metallic bond structure, which generates an electrostatic force of attraction due to the positive titanium atoms. The main amalgam phase, the γ-1 phase, is formed on the (121) crystallographic plane [25] and is observed to adhere to the titanium in trace amounts [26]. More titanium is observed to be attached to the eta-phase, thus contributing to the hardness properties of the alloy [42]. The addition of heat to the amalgam would further degrade the structure and would render the material ineffective for use in the oral environment. The preservation of the material was significant, and such renders the results more precise. Titanium is considered biocompatible with human tissues; therefore, its effect on the human immune system is not critical [43]. It is not well known how the orientation of the crystallographic (121) γ-1 plane affects the titanium nanoparticles doped amalgam, but in accordance with the results obtained, each titanium nanoparticles doped amalgam sample indicated a significant decrease in mercury vapor levels when measured using the Jerome J505 Mercury Vapor Analyzer instrument.

Figure 2a,b shows the release of Hg vapor from the Sybralloy (33% Cu) dental amalgam sample before the addition of titanium-doped particles to the alloy. Without the addition of titanium, the plot shows that the Hg vapor concentration measured (black line) is significantly high. When 10 mg of titanium particles were added to the sample, the Hg vapor released showed a decrease in Hg vapor (red Line). Subsequently, adding 20 mg of titanium to the sample further decreased the Hg vapor (blue line). Finally, with the addition of 30 mg titanium (green line), a further decrease in Hg vapor was observed, as seen in Figure 2. Based on the plot of Figure 2, the released Hg vapor pattern shows a near uniformity in the pattern of decreased Hg vapor. The micro-mechanism of such phenomena in reducing the Hg vapor from the material results from the titanium atoms displacing the tin in the γ-2 phase, thus depressing the release of Hg vapor [20]. The research demonstrated that the Sybralloy dental amalgam, having 33% copper within its composition, showed a decrease in mercury vapor concentration, the release of which is primarily from the grain boundaries (as would be seen in the SEM/EDS images subsequently) of the various phases formed subsequent to condensation. The material grain boundaries form the barrier between the elemental mercury and the phases formed within the matrix of the triturated material; most importantly, mercury vapor is released primarily in the γ_1_-phase (Ag_2_ Hg_3_), which is the main matrix phase of the triturated material. The addition of titanium to the Sybralloy (33% Cu) further reduces the mercury vapor and further increases its hardness properties, together with increasing its biocompatibility. Measurements were taken over a 400 s duration of data acquisition. Choosing the range of 400 s is of no particular significance.

The regression and ANOVA analysis, as seen in Table 5 and Table 6, showed a very good R^2^ (coefficient of determination) factor, which was greater than 97% in all cases, and a very good model fit (ref. Figure 1b and Figure 2b), which can then be extrapolated to events after 400 s, with a high degree of freedom. This further confirms the validity of these extrapolations. The slopes of the fitted lines in Table 5 and Table 6 also show a not-so-rapid decline in the rate of mercury vapor released, as it ranges from −0.133 to −0.693.

Figure 3 shows the comparison of the Vickers hardness between the Dispersalloy (11.8% Cu) amalgam and the Sybralloy (33% Cu) amalgam. According to the graph displayed, the Hg vapor released from the Dispersalloy prior to the addition of titanium was approximately 15,000 kg/m^3^ after 400 s, as observed in Table 2, and the Hg vapor released from the Sybralloy prior to the addition of titanium is about 10,000 kg/m^3^. With the addition of 10 mg, 20 mg, and 30 mg titanium to both the Dispersalloy dental amalgam brand and the Sybralloy dental amalgam brand, the graph shows the amount of Hg vapor released from the Sybralloy amalgam is slightly less than 10,000 kg/m^3^. Based on prior research data on the Hg vapor release measurements, the Hg vapor released over a wider range of time would not significantly deviate from the pattern of vapor released, as shown in Figure 3.

The Sybralloy (33% Cu) is categorized as a high-copper dental amalgam. This brand of dental amalgam tends to release very little Hg vapor in comparison to the other amalgam brands currently on the market that are being utilized by dentists. The 33% copper addition to the material inhibits the release of large amounts of Hg vapor; the measured data of the results can be observed in Table 3. The addition of titanium to the Sybralloy amalgam brand further suppresses the Hg atoms from the main matrix and further increases the hardness properties of the alloy. As shown in Figure 3, the total mercury vapor level released from both the Dispersalloy (11.8% Cu) brand of amalgam and the Sybralloy (33% Cu) brand of amalgam shows an almost equal amount of Hg vapor released for both brands (Dispersalloy and Sybralloy) of dental amalgam, with and without the addition of titanium particles. The average concentration difference in total Hg vapor released between the Dispersalloy amalgam and the Sybralloy amalgam is about 5000 kg/m^3^. This justification in close uniform distribution is not readily apparent but can be construed due to its average uniform distribution of titanium atoms within the major γ-1 phase of the alloy.

Figure 4 shows the titanium nanoparticles interspersed among the Hg atoms, thus aiding in the elimination of the Sn_8_Hg phase. The adhering of titanium atoms within the metal matrix displays the reddish-brown pigmentation in the above SEM image. The trace amount of titanium is observed to combine with the copper Cu_6_Sn phase of the alloy, which further strengthens the material, together with the copper enhancement of the eta-phase.

Both the copper and titanium in the matrix of the alloy contribute to the hardening of the alloy, likewise decreasing creep and decreasing the mercury vapor that is released from the amalgam. The trace amount of titanium is noticeable within the silver and copper particles, but the actual mechanism of binding is not well understood. It is expected that the amount of mercury vapor released from the amalgam prior to the addition of titanium will be undetermined until the measurements are obtained. The addition of 10 mg of titanium did effectively reduce the quantity of mercury vapor linearly, followed by the addition of 20 mg and 30 mg of titanium.

Figure 5 shows the SEM image of the Sybralloy amalgam consisting of the addition of 20 mg titanium. The titanium nanoparticles are interwoven among the copper atoms, which indicates that the eta-phase consists of some trace amount of titanium particles. The addition of the titanium particles indeed serves a dual purpose, which is (1) additional strength to the material and (2) additional eliminations of Hg vapor from the γ-1 and, if possible, the γ-2 phase [42].

Figure 5, in addition, demonstrates an increased area of titanium mixed metallic phases, which is noticed from the light brown-colored portion (a mixture of the titanium and mercury) seen from the image. Adding 20 mg of titanium to the Sybralloy (33% Cu) resulted in a much further reduction in mercury vapor from this brand of amalgam. The reduction in mercury vapor from the amalgam is independent of the brand or type of amalgam since all amalgams are manufactured with the same fundamental building blocks, which are silver, tin, copper, zinc, mercury, and a trace amount of indium and palladium. Almost a linear pattern of release in the Hg vapor is noticeable, as is observed in Figure 1 and Figure 2. Such pattern of Hg vapor released demonstrated a nearly linear pattern of vapor decreased in consideration to that of a negative slope formation, which was noticeable when the 10 mg, 20 mg, and 30 mg were added to the Dispersalloy (11.8% Cu) and the Sybralloy (33% Cu).

Figure 5 shows the SEM image for the doping of the Dispersalloy amalgam with 30 mg titanium. The image from the SEM shows that the titanium nanoparticles are also finely distributed among the Hg and copper atoms (brownish color). This arrangement and distribution of titanium atoms further serves as the mechanism for reducing the Hg vapor by displacing some of the Hg atoms from the γ_1_-phase (Ag_2_Hg_3_) while simultaneously improving the hardness properties of the material by strengthening the alloy (as will be seen in the Vickers hardness plot later). The addition of the 30 mg titanium was observed to show a wider dispersion of titanium interspersed among the various phases formed by the material subsequent to trituration. Titanium could be noticed to have a high affinity for the major matrix phase, the γ-1 phase (Ag_2_Hg_3_). The excess addition of titanium to the amalgam would not be effective for clinical usage due to rendering the material excessively hard (high creep), and such phenomena would create brittleness within the grain boundaries of the material. A reduction in clinical working time would significantly increase, and an inadequate adaptation of the material to the teeth’s internal structure would become noticeable. Figure 4 and Figure 6 show some slight similarities in the distribution of titanium atoms within the main matrix of the amalgam. The image in Figure 7a–f also confirms the fairly even distribution of Hg, Ag, Sn, Cu, Ti, and Zn.

Figure 8 shows the increase in Vickers hardness for all eight samples of the Dispersalloy (11.8% Cu), starting with the first sample having 10 mg titanium, followed by the second sample having 20 mg titanium, the third sample having 30 mg titanium, the fourth sample having 40 mg titanium, the fifth sample having 50 mg titanium, the sixth sample having 60 mg titanium, the seventh sample having 70 mg titanium, and the eighth sample having 80 mg titanium. According to the representation in the graph of Figure 8, the increase in 10 mg weight percent titanium in each Dispersalloy amalgam sample thus contributes to an increased Vickers hardness for each sample, as is observed on the graph. Other properties of the material can be inferred to be improved by the addition of titanium, such as clinical creep, longevity, biocompatibility, and reduced oxidation formation on the surface of the material, although this was not analyzed in this paper.

The XRD image in Figure 9 shows the combinations of the metallic element constituents of the improved dental amalgam with the addition of titanium particles. As observed, the titanium particles were absorbed among the γ-1 phase and possibly the γ-2 phase, which creates a reduction in the mercury vapor released. The precise behavioral mechanism of this phenomenon is still unknown. Further detailed research on this topic would be required to address the fundamental bonding mechanism of the titanium with the other phases in the alloy.

## 4. Conclusions

The present research demonstrated that an addition of a small amount of Ti nanoparticles to mercury-based dental amalgams can result in a considerable decrease in Hg vapor release. The decrease in the Hg vapor release due to Ti addition has been explained by the formation of strong Hg–Ti covalent bonds, which reduce the availability of Hg atoms for evaporation. The Ti atoms more than the solubility limit of Ti in Hg reside in the grain boundaries, which also reduce the evaporation of Hg from the amalgam. Simultaneously, with the addition of titanium to the dental amalgam in the present research, the hardness properties of the alloy were significantly improved while decreasing the mercury vapor levels from the amalgam. Changes in material properties were observed. Upon the trituration of the amalgam alloy, mercury diffuses into the silver–tin phase, or the γ-phase, γ. The titanium atoms accumulate on the external surface of this γ-phase (Ag_3_Sn). Some of the titanium atoms present in the mixture migrate to the γ-2 phase to combine with the tin in the Sn_8_Hg phase. The titanium is known to displace the tin in the γ-2 phase. Incorporating titanium nanoparticles within the matrix of the amalgam alloy alters the hardness properties of the amalgam and can be inferred to decrease creep, prolong the clinical duration of the amalgam, and reduce the oxidation of the material. The excess addition of titanium to the amalgam would not be effective in clinical usage due to rendering the material excessively hard (high creep), and such phenomena would create brittleness within the grain boundaries of the material. A reduction in clinical working time would significantly increase, and an inadequate adaptation of the material to the teeth’s internal structure would become noticeable.

## Figures and Tables

**Figure 1 materials-17-01662-f001:**
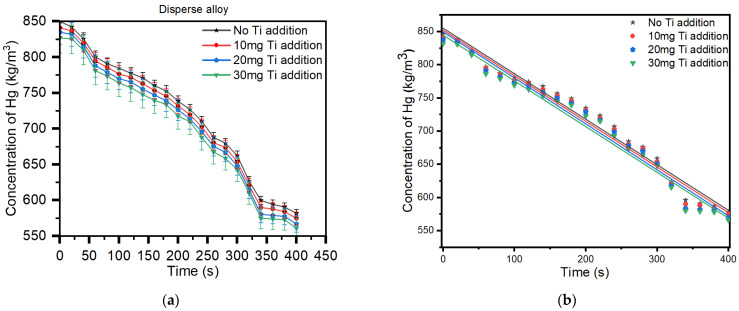
(**a**) Mercury vapor released from the Dispersalloy (11.8% Cu) before and after the addition of 10 mg Ti, 20 mg Ti, and 30 mg Ti. (**b**) The linear regression analysis of the plots.

**Figure 2 materials-17-01662-f002:**
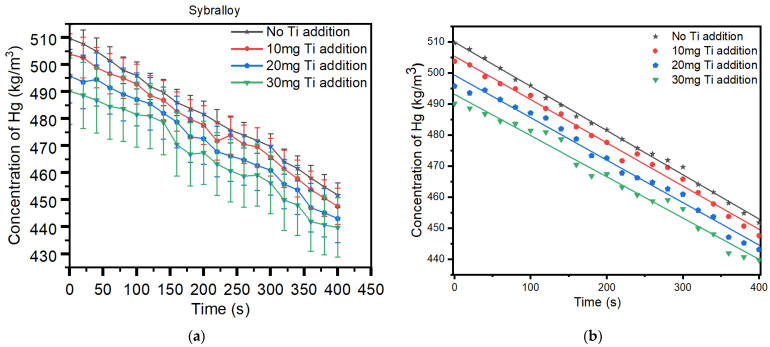
**(a)** Mercury vapor released from the Sybralloy (33% Cu) before and after the addition of 10 mg Ti, 20 mg Ti, and 30 mg Ti. (**b**) The linear regression analysis of the plots.

**Figure 3 materials-17-01662-f003:**
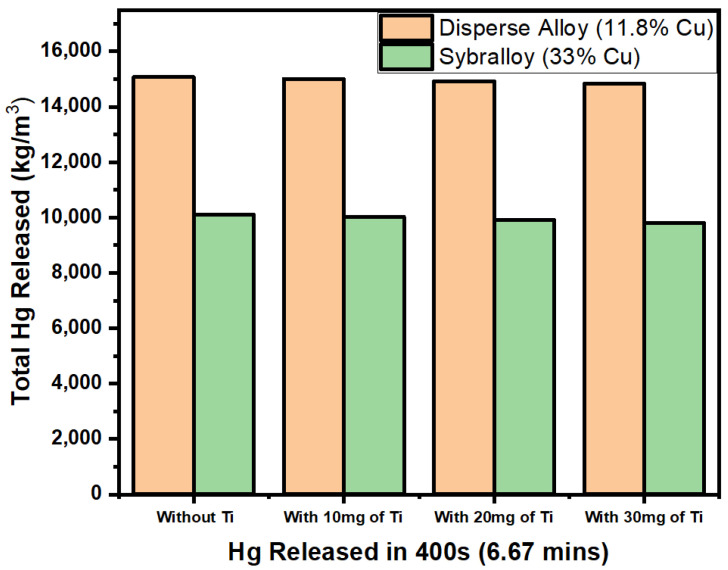
Hg vapor released from the Dispersalloy (11.8% Cu) and the Sybralloy (33% Cu) dental amalgam without titanium-doped particles and with 10 mg, 20 mg, and 30 mg titanium-doped particles.

**Figure 4 materials-17-01662-f004:**
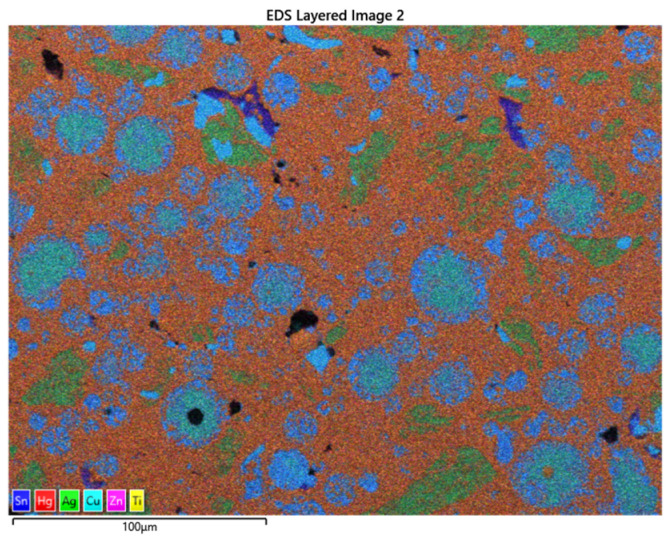
SEM/EDS of the Dispersalloy amalgam sample with the addition of 10 mg titanium nanoparticles.

**Figure 5 materials-17-01662-f005:**
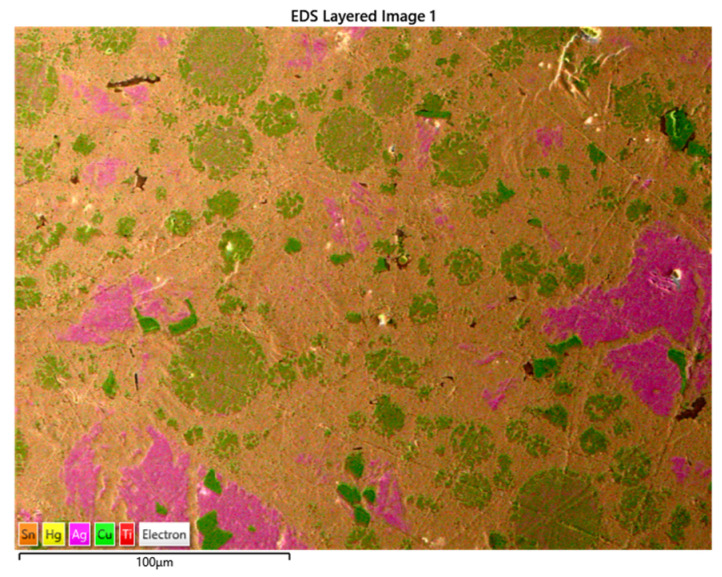
SEM of the Sybralloy amalgam sample with the addition of 20 mg titanium-doped particles.

**Figure 6 materials-17-01662-f006:**
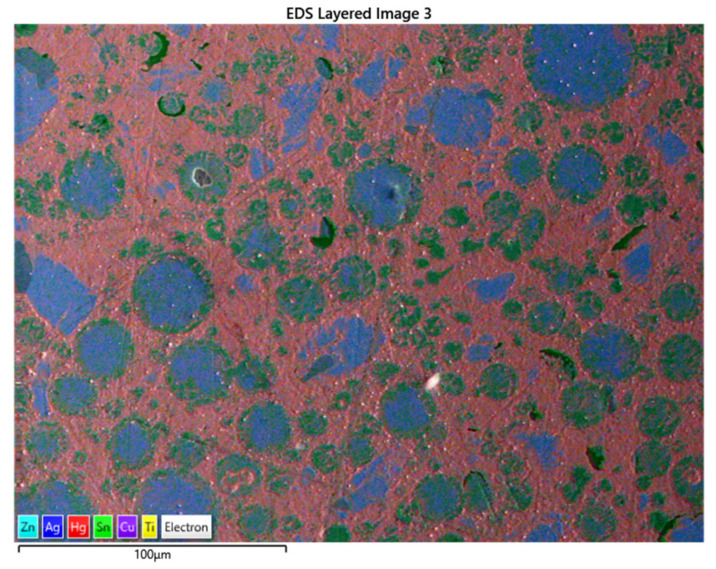
SEM of the Dispersalloy amalgam sample with the addition of 30 mg titanium-doped particles.

**Figure 7 materials-17-01662-f007:**
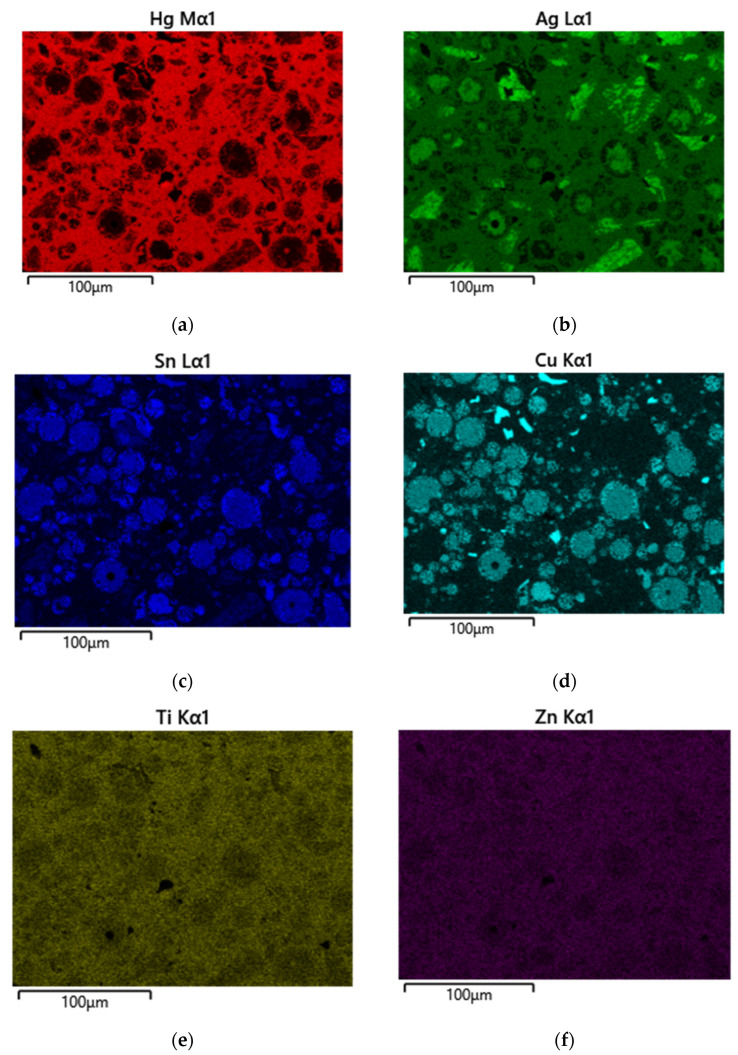
(**a**) Hg, (**b**) Ag, (**c**) Sn, (**d**) Cu, (**e**) Ti, and (**f**) Zn indicate the metallic constituents present in the amalgam alloy subsequent to trituration in the Dispersalloy amalgam.

**Figure 8 materials-17-01662-f008:**
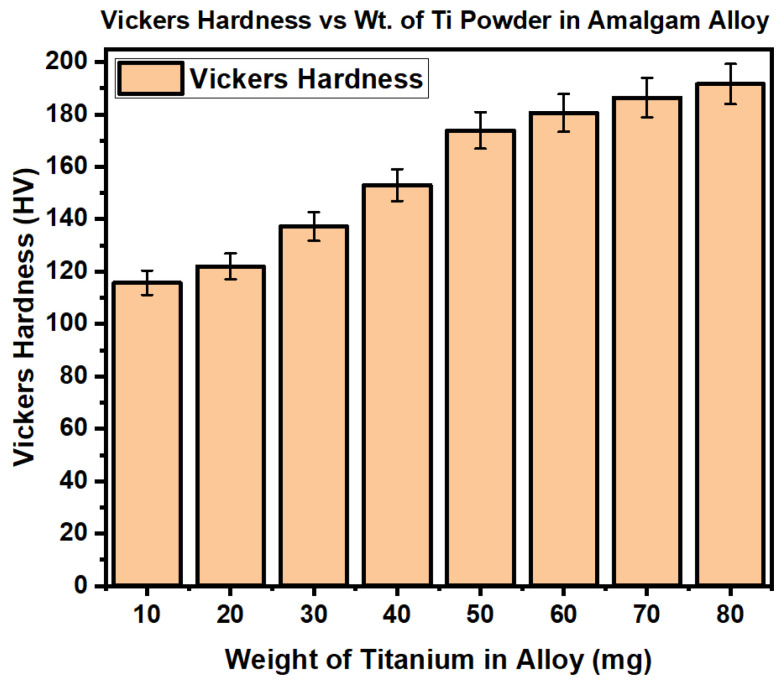
Increase in Vickers hardness for the eight samples of the Dispersalloy (11.8% Cu) dental amalgam versus the increase in weight percent titanium in the alloy with error bar.

**Figure 9 materials-17-01662-f009:**
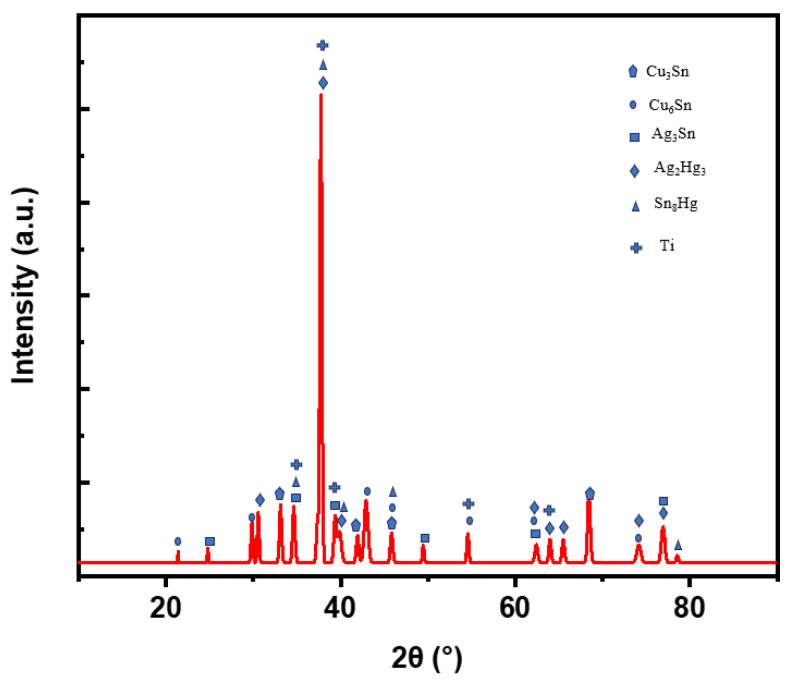
XRD for the Dispersalloy amalgam with 10 mg titanium nanoparticles within the matrix. γ-1—dominant matrix phase (Ag_2_Hg_3_), γ-2—corrosive phase (Sn_8_Hg), β-1—dendrite phase (Ag_3_Sn), η-1—unreacted nucleation and protection phase (Cu_3_Sn), and η-2 1—reacted nucleation and protection phase (Cu_6_Sn).

**Table 1 materials-17-01662-t001:** Amalgam composition before and after the addition of Hg.

Amalgam	Silver	Tin	Copper	Zinc	Mercury
Dispersalloy (Before)	69.5%	17.70%	11.80%	1.00%	-
(After)	34.75%	8.85%	5.90%	0.50%	50.00%
Sybralloy (Before)	40.00%	26.80%	33.00%	0.20%	-
(After)	18.4%	12.33%	15.18%	0.09%	46.00%

**Table 2 materials-17-01662-t002:** Table showing the concentration of Hg vapor released from the Dispersalloy dental amalgam without the addition of titanium and the Hg vapor released from the amalgam when 10 mg, 20 mg, and 30 mg of titanium are added to the Dispersalloy amalgam.

Time (s)	Disperse Alloy without Titanium (kg/m^3^)	10 mg Ti (kg/m^3^)	20 mg Ti (kg/m^3^)	30 mg Ti (kg/m^3^)
0	846.085	840.912	837.684	831.746
20	838.5543	836.753	834.506	830.610
40	821.8564	819.484	817.615	814.582
60	796.6544	794.553	790.832	786.064
80	787.0785	785.748	781.475	778.509
100	780.6533	776.378	772.671	768.917
120	773.851	771.637	768.083	762.455
140	767.3113	762.788	758.149	752.788
160	756.4582	753.443	749.743	744.506
180	748.9329	745.518	741.826	738.884
200	734.831	731.667	728.925	722.617
220	723.0086	719.471	716.337	714.841
240	706.5386	702.661	698.586	692.789
260	684.1459	680.305	677.990	672.447
280	675.8653	673.548	669.812	663.599
300	658.6462	653.709	650.716	647.443
320	623.0678	620.806	617.448	614.909
340	595.9563	589.642	583.067	579.712
360	590.5342	587.812	581.549	578.801
380	586.6308	583.904	580.172	577.759
400	577.8899	574.817	569.553	565.762

**Table 3 materials-17-01662-t003:** Table showing the amount of Hg vapor released from the Sybralloy amalgam without the addition of titanium and the Hg vapor released when 10 mg, 20 mg, and 30 mg of titanium were added to the Sybralloy amalgam.

Time (s)	Sybralloy without Titanium (kg/m^3^)	10 mg Ti (kg/m^3^)	20 mg Ti (kg/m^3^)	30 mg Ti (kg/m^3^)
0	509.658	503.772	495.746	490.128
20	507.684	502.607	493.581	488.607
40	504.855	498.818	494.473	486.790
60	501.534	496.573	491.455	484.482
80	497.867	494.914	488.964	483.617
100	495.953	492.763	487.109	481.448
120	491.894	488.548	485.476	480.891
140	489.657	486.817	481.990	478.609
160	485.967	482.636	478.754	470.445
180	483.746	479.837	473.305	466.781
200	481.676	477.649	472.568	467.408
220	478.656	471.715	467.783	463.172
240	475.767	473.883	466.191	460.688
260	473.866	470.499	464.736	458.711
280	471.860	469.565	462.643	459.073
300	469.699	465.772	460.908	456.189
320	464.076	461.439	455.757	449.909
340	461.568	457.808	453.664	448.055
360	458.063	453.733	447.067	441.927
380	454.768	450.614	445.196	440.675
400	451.755	447.547	443.063	439.742

**Table 4 materials-17-01662-t004:** Total mercury vapor concentration released for each of the Dispersalloy and the Sybralloy amalgams without the addition of titanium and for the addition of 10 mg, 20 mg, and 30 mg of titanium.

	Disperse Alloy—11.8% Cu (kg/m^3^)	Sybralloy—33% Cu (kg/m^3^)
Total Concentration of Hg Without Titanium	15,074.5499	10,110.569
Total Concentration of Hg with 10 mg of Titanium	15,005.556	10,027.509
Total Concentration of Hg with 20 mg of Titanium	14,926.739	9910.429
Total Concentration of Hg with 30 mg of Titanium	14,839.74	9797.347

**Table 5 materials-17-01662-t005:** Analysis of variance (ANOVA) for the different Dispersalloy with various titanium contents.

	Slope		DF	Sum of Squares	Mean Square	F Value	Prob > F
Disperse Alloy Without Titanium (R^2^ = 0.97621)	−0.686	Model	1	144,988.2712	144,988.2712	779.48714	1.11 × 10^−16^
		Error	19	3534.08927	186.0047		
		Total	20	148,522.3605			
10 mg Ti (R^2^ = 0.97626)	−0.688	Model	1	145,777.7609	145,777.7609	781.45524	1.11 × 10^−16^
		Error	19	3544.38401	186.54653		
		Total	20	149,322.1449			
20 mg Ti (R^2^ = 0.97459)	−0.693	Model	1	147,814.1458	147,814.1458	728.70775	1.11 × 10^−16^
		Error	19	3854.03994	202.84421		
		Total	20	151,668.1857			
30 mg Ti (R^2^ = 0.97563)	−0.689	Model	1	146,448.5597	146,448.5597	760.55004	1.11 × 10^−16^
		Error	19	3658.56614	192.55611		
		Total	20	150,107.1259			

**Table 6 materials-17-01662-t006:** Analysis of variance (ANOVA) for the different Sybralloy with various titanium contents.

	Slope		DF	Sum of Squares	Mean Square	F Value	Prob > F
Sybralloy Without Titanium (R^2^ = 0.99721)	−0.143	Model	1	6265.08552	6265.08552	6802.26144	0
		Error	19	17.49957	0.92103		
		Total	20	6282.58509			
10 mg Ti (R^2^ = 0.99216)	−0.14	Model	1	5995.10186	5995.10186	2405.08619	0
		Error	19	47.36085	2.49268		
		Total	20	6042.46271			
20 mg Ti (R^2^ = 0.98835)	−0.137	Model	1	5782.32621	5782.32621	1611.21658	0
		Error	19	68.18711	3.5888		
		Total	20	5850.51332			
30 mg Ti (R^2^ = 0.9834)	−0.133	Model	1	5445.9406	5445.9406	1125.60529	0
		Error	19	91.92643	4.83823		
		Total	20	5537.86702			

## Data Availability

Data is contained within the article.

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
