# Peer review of "Dental Metal Matrix Composites: The Effects of the Addition of Titanium Nanoparticle Particles on Dental Amalgam"

_materials, 2024, doi:10.3390/ma17071662_

Round 1

Reviewer 1 Report

Comments and Suggestions for Authors

The manuscript describes the investigation of the effect of addition of titanium powder to two brands of dental amalgam on the properties, such as mercury vapor concentration, phase changes and hardness.

The topic and scope suit the Topic “Metal Matrix Composites: Recent Advancements”; the methods are appropriate and the results support the conclusion. Title typo “doped” must be corrected.

The overall work needs major revision in improving problem statement, hypothesis and objectives introduction and result dissemination.

The Abstract needs to be rewritten to reduce general introduction and to state the objectives and hypothesis, main results, conclusion and significance. “creep and corrosive resistance” have not been studied thus should not be mentioned.

The Introduction must be reformulated to introduce the general literature, problem statement, what specific objectives the work was designed to address. The current version appears to be an extensive summary.

Scientific rigor must be improved, the addition of Ti powder should be introduced first rather than at the end. Statistical analysis has not been done to verify if any difference is statistically significant. Without this, no rational conclusions can be drawn.

The result dissemination is most problematic:

1. Many figures are not mentioned in section 4, hence result discussion is not effective but confusing.

2. is the vapor measurement only for the first 400 s? what about afterwards?

3. although linear pattern has been mentioned in places, no linear regression has been done.

4. in some Figures, y-axis label is not consistent with caption.

5. Fig. 4-8, the captions do not specify which sample results. Also please add pointers to guide the eye.

6. Lots of speculations have been made without evidence, such as Line 233-238, 276-279.

7. Fig. 9 should be moved forward to after figure 2 since these are all Hg release results.

8. Fig. 10, please specify which sample, how much Ti? Should also add gamma-1, gamma-2 symbols, etc.

9. Conclusion should be a separate section and comprehensive. Please add clinical significance.

When introduce amalgam in general and the phase composition, suggest citing a work that explicitly explores this, Geometric, electronic and elastic properties of dental silver amalgam γ-(Ag3Sn), γ1-(Ag2Hg3), γ2-(Sn8Hg) phases, comparison of experiment and theory, Intermetallics 2010.

Author Response

The Abstract needs to be rewritten to reduce general introduction and to state the objectives and hypothesis, main results, conclusion and significance. “creep and corrosive resistance” have not been studied thus should not be mentioned.

Response: The Abstract has been rewritten incorporating the referee’s suggestions. The general introduction text has been removed and objective of the present research has been clearly stated now.

The Introduction must be reformulated to introduce the general literature, problem statement, what specific objectives the work was designed to address. The current version appears to be an extensive summary.

Response: This has been carried out, the general introduction and problem statement are 1st reported followed by objectives, an extensive literature review of past research in this field.

Scientific rigor must be improved, the addition of Ti powder should be introduced first rather than at the end. Statistical analysis has not been done to verify if any difference is statistically significant. Without this, no rational conclusions can be drawn.

Response: The rationale of the method has been discussed with reference to previous studies to substantiate some of the results made in the

The result dissemination is most problematic:

Response: This has been addressed in the revised manuscripts all result discussion is limited to the abstract, results & discussion section and the conclusion.

  1. Many figures are not mentioned in section 4, hence result discussion is not effective but confusing.

Response: All figures & tables have now been discussed effectively and highlighted in bold for emphasis.

  1. is the vapor measurement only for the first 400 s? what about afterwards?

Response: Measurements were taken over a 400 second duration of data acquisition. Choosing the range of 400 seconds is of no particular significance as the R2 factor will prove the linear model is very accurate for predicting the future vapor release in the near hours. Also, with a high degree of Freedom this further confirms the validity of these extrapolations. The slopes of the fitted lines in Table 4 and 5 also show a not so rapid decline in the rate of mercury vapor released as it ranges from -0.133 to -0.693.

  1. although linear pattern has been mentioned in places, no linear regression has been done.

Response: This has been done see Table 4 and 5 and see fig 1b & 2b.

  1. in some Figures, y-axis label is not consistent with caption.

Response: This has been removed/addressed.

  1. Fig. 4-8, the captions do not specify which sample results. Also please add pointers to guide the eye.

Response:This has been added, Thank you. See revised manuscript.

  1. Lots of speculations have been made without evidence, such as Line 233-238, 276-279.

Response: This has been addressed and substantiated with appropriate references.

  1. Fig. 9 should be moved forward to after figure 2 since these are all Hg release results.

Response: This has been moved upwards as has been suggested.

  1. Fig. 10, please specify which sample, how much Ti? Should also add gamma-1, gamma-2 symbols, etc.

Response: This has been specified as dispersalloy amalgam with 10mg Titanium. See the figure description for the needed references.

  1. Conclusion should be a separate section and comprehensive. Please add clinical significance.

Response: A new section on conclusion has been included and text has been rewritten.

Reviewer 2 Report

Comments and Suggestions for Authors

Abstract

- Please keep the abstract format according to the journal’s instructions and refer to the methods utilized here.

 In the present work an effort was made to reduce the Hg vapor release from dental amalgams along with hardness increase, by adding titanium particles. After my consideration, I would suggest the following issues to be addressed: 

Introduction

-The presentation of the study purpose is duplicated here. Provide the purpose clearly in the last paragraph.

-The obtained results should not be included in this section.

- You should highlight the importance of your work in the last paragraph.

2. Materials and Equipment / 3. Methodology

- Both sections should be entitled as Materials and methods according to journal’s guidelines.

- Provide information about the origin of used titanium powder (e.g. purity, particle average size, manufacturer) .

- Give the detailed composition of the studied amalgams in a separate Table here, other than in the introduction.

- Describe the SEM and Vickers hardness instrumentation in details.

-Lines 189-204: The described information does not belong to the experimental procedure and should be avoided.

- The last paragraph should be merged with the first paragraph.

- Regarding the determination of the Hg vapor released concentration, how many replicates did you perform? This issue is of high importance and should be highlighted here.

Results

- Figures 1 & 8 describe the addition of Titanium over 30 mg, while nowhere else in the text this is apparent. A detailed explanation should be given about this issue.

-Tables 1, 2 & 3. Mean values accompanied with standard deviations should be presented. Reduce the decimal digits to 2. Did you use any statistical method to compare the obtained values?

-Tables 1 & 2: Refer to the % Cu once (e.g. in legend) for each amalgam type, instead of the first Table line.

-Table 3. Metric units are missing. The Table should be placed close to the discussed text content.

- The majority of Figures is not mentioned in the text.

- Which amalgam is represented in Figure 8? The corresponding Vickers hardness results for the second amalgam are missing.

Author Response

Referee-2

 Please keep the abstract format according to the journal’s instructions and refer to the methods utilized here.

Response: This has been addressed in the revised manuscripts.

 In the present work an effort was made to reduce the Hg vapor release from dental amalgams along with hardness increase, by adding titanium particles. After my consideration, I would suggest the following issues to be addressed: 

Introduction

-The presentation of the study purpose is duplicated here. Provide the purpose clearly in the last paragraph.

Response: This has been edited and moved completely to the 1st paragraph.

-The obtained results should not be included in this section.

Response: This has been removed and addressed.

- You should highlight the importance of your work in the last paragraph.

 Response: This has been done, see line 35-44 of the revised manuscripts.

  1. Materials and Equipment / 3. Methodology

- Both sections should be entitled as Materials and methods according to journal’s guidelines.

Response: The 2 sections have been merged as “Materials and Methods”.

- Provide information about the origin of used titanium powder (e.g. purity, particle average size, manufacturer).

Response: This has been done see lines 151-156 of the corrected manuscript.

- Give the detailed composition of the studied amalgams in a separate Table here, other than in the introduction.

Response: This has been done see table a in the Materials and Methods.

- Describe the SEM and Vickers hardness instrumentation in details.

Response: This has been done for SEM in lines 189-192 & Vickers hardness instrumentation lines 195-212.

-Lines 189-204: The described information does not belong to the experimental procedure and should be avoided.

Response: This has been addressed and moved to the introduction.

- The last paragraph should be merged with the first paragraph.

Response: This has been addressed in the revised manuscripts.

- Regarding the determination of the Hg vapor released concentration, how many replicates did you perform? This issue is of high importance and should be highlighted here.

 Response: This has been addressed in the revised manuscripts lines 201-203.

Results

- Figures 1 & 8 describe the addition of Titanium over 30 mg, while nowhere else in the text this is apparent. A detailed explanation should be given about this issue.

Response: This has been extensively addressed in the revised manuscripts all. It is also discussed in the Materials and Method. All data is provided as convenient and not to over suffocate the plots with too many data points when a few data points can still tell the same story

-Tables 1, 2 & 3. Mean values accompanied with standard deviations should be presented. Reduce the decimal digits to 2. Did you use any statistical method to compare the obtained values?

Response: The table 3 refers to the sum of the

-Tables 1 & 2: Refer to the % Cu once (e.g. in legend) for each amalgam type, instead of the first Table line.

Response: This has been removed to avoid confusion.

-Table 3. Metric units are missing. The Table should be placed close to the discussed text content.

Response: This has been added.

- The majority of Figures is not mentioned in the text.

Response: This has been addressed and all figures and tables in the revised manuscripts are discussed in detail.  

- Which amalgam is represented in Figure 8? The corresponding Vickers hardness results for the second amalgam are missing.

Response: It is the dispersalloy, it has been updated see revised manuscripts.

Round 2

Reviewer 1 Report

Comments and Suggestions for Authors

The authors have addressed the majority of the points raised, except one important point "Statistical analysis has not been done to verify if any difference is statistically significant. Without this, no rational conclusions can be drawn."

All measurements should be repeated for at least 3 times and the results should be presented as mean (standard deviation), and in the result plots with error bars representing standard deviation. Statistical analysis varifies if the difference between groups is statistically significant.

Without this, "significant(ly)" can not be used justifiably, no rational conclusions can be drawn.

Please double check ref [35], one author appears twice.

Author Response

The authors have addressed the majority of the points raised, except one important point "Statistical analysis has not been done to verify if any difference is statistically significant. Without this, no rational conclusions can be drawn."

All measurements should be repeated for at least 3 times and the results should be presented as mean (standard deviation), and in the result plots with error bars representing standard deviation. Statistical analysis varifies if the difference between groups is statistically significant.

Without this, "significant(ly)" can not be used justifiably, no rational conclusions can be drawn.

Response: This was addressed earlier in lines 201-203 where it was stated that the experimental data was established as an average of data points achieved under identical conditions (at least 4 times). An analysis of the variance (ANOVA) was carried out by taking a mean of the 4 different amalgam conditions with respect to the titanium content; the best fit for the regression model showed good R2 factor (≥97%); see lines 315-320. The error bars have been included in Fig. 1a, 2a and 8.

Also, the word ‘significant’ in the abstract has been replaced with ‘considerable’ in lack of a rigorous statistical analysis.

Please double check ref [35], one author appears twice.

Response: Thank you for the comment. This has been addressed.

Reviewer 2 Report

Comments and Suggestions for Authors

Dear Authors,

I would suggest the revised version for publication.

Author Response

NA